

# Measurement Report: Impact of African Aerosol Particles on Cloud Evolution in a Tropical Montane Cloud Forest in the Caribbean

Elvis Torres-Delgado[1], Darrel Baumgardner[2], Olga L. Mayol-Bracero[1]

[1]Department of Environmental Science, University of Puerto Rico, San Juan, 00925, Puerto Rico
[2]Droplet Measurement Technologies, Longmont, CO, 80503, USA

*Correspondence to*: Elvis Torres-Delgado (elvis.torres810@gmail.com)

**Abstract** African aerosol particles, traveling thousands of kilometers before reaching the Americas and the Caribbean, directly scatter and absorb solar radiation and indirectly impact climate by serving as cloud condensation nuclei (CCN) or ice nuclei (IN) that form clouds. These particles can also affect the water budget by altering precipitation patterns that subsequently affect ecosystems. As part of the NSF-funded Luquillo Critical Zone Observatory, field campaigns were conducted during the summers of 2013, 2014, and 2015 at Pico del Este, a site in a tropical montane cloud forest on the Caribbean island of Puerto Rico. Cloud microphysical properties, which included liquid water content, droplet number concentration, and droplet size, were measured. Using products from models and satellites, as well as in-situ measurements of aerosol optical properties, periods of high and low dust influence were identified. The results from this study suggest that meteorology and air mass history have a more important effect on cloud processes than aerosols transported from Africa. In contrast, air masses that arrived after passing over the inhabited islands to the southeast led to clouds with much higher droplet concentrations, presumably due to aerosols formed from anthropogenic emissions.

## 1 Introduction

Great strides have been made in our understanding of aerosol-cloud-radiation interactions; however, substantial uncertainty remains (Mhyre et al., 2013), particularly in our understanding of the role of mineral dust during such interactions. Dust has a direct impact on climate since it is considered one of the most important light-absorbing particles in the atmosphere at shorter solar wavelengths (Sokolik and Toon, 1999). Dust is a good ice nucleus (DeMott et al., 2003), but its activity as cloud condensation nuclei (CCN) is less clear. Studies have shown that fresh dust is mainly hydrophobic and a poor CCN (Rosenfeld et al., 2001) while others have shown that dust can act as a CCN if it has aged and mixed with anthropogenic or organic aerosols (Fitzgerald et al., 2015). The later observation is supported by laboratory studies that have found that the CCN activity depends on the mineralogical composition and mixing state of the dust particle (Sullivan et al., 2009; Gaston, 2020). The larger, super-micron dust particles act as giant CCN that can form large cloud droplets that lead to the earlier formation of raindrops (Rosenfeld et al., 2008).

In the Caribbean Basin, which is frequently inundated by mineral dust picked up by surface winds over the northern half of the African continent and transported westward, studies of aerosol-cloud-radiation interactions are





limited with most having been done on the islands of Puerto Rico and Barbados. Within the northeastern part of Puerto Rico is a tropical montane cloud forest (TMCF) known as Pico del Este (PDE). PDE has been an ideal location for studying clouds as they form frequently and last for many hours. Eugster et al. (2006) studied daily cloud properties

as a function of net radiation and saw that denser clouds correlated with less radiation at the surface. Allan et al. (2008) observed that clouds affected by anthropogenic pollution had a larger number of droplets with smaller diameters. The results in Spiegel et al. (2014) support the observations in Allan et al. (2008) but with clouds impacted by African dust. In contrast, Raga et al. (2016) observed that meteorology was more important in cloud formation than the aerosol source, particularly during dust events. Cloud chemical properties have also been studied at this location (Weathers et

al., 1988; Asbury et al., 1994; Gioda et al., 2008, 2009, 2011, 2013; Reyes-Rodríguez et al., 2009; Valle-Díaz et al., 2016). Under the influence of African dust, there was an increase in non-sea salt calcium and other crustal and organic components when clouds were formed during periods of African dust (Valle-Díaz et al., 2016). There was also an increase in non-sea salt sulfate when under the influence of anthropogenic pollution (Gioda et al., 2009; Valle-Díaz et al., 2016).


The main objective of the study being reported here is to use measurements of cloud microphysical properties in a TMCF, complemented with aerosol measurements and back trajectory analysis, to evaluate the relationship between the source of aerosols and the formation and evolution of clouds. The working null hypothesis is that at this location, the source of the aerosols will have a statistically insignificant effect on the clouds properties.


## 2 Methodology

### 2.1 Sampling Sites

Measurements were made at coastal and mountain sites on the Caribbean island of Puerto Rico. The coastal

site is an observatory within the nature reserve of Cabezas de San Juan (CSJ, 18° 23'N, 65°37'W), in the most northeastern part of Puerto Rico near the town of Fajardo. The station experiences northeasterly trade winds most of the time, which makes it an ideal place to study background, marine aerosols as the site is located upwind of the metropolitan area, a potential source of anthropogenic particles. The trade winds can bring particles from the desert in Africa, a common occurrence during the Northern Hemisphere summer (Prospero & Mayol-Bracero, 2013).

Anthropogenic pollution is also transported from North America by cold fronts, more commonly during the Northern Hemisphere winter, and also from the nearby islands during southeasterly winds (Allan et al., 2008; Gioda et al., 2009; Valle-Díaz et al., 2016).

The mountain station is located at the Pico del Este (PDE, 18°16' N, 65°45' W), an observatory in the tropical

montane cloud forest located within the El Yunque National Forest at an elevation of 1051 m asl. PDE is often engulfed in long-lived clouds, which makes it a useful place to study cloud properties over extended periods of time. This station is also upwind of the city of San Juan, a main metropolitan area, thus rarely influenced by local anthropogenic pollution, but downwind from the CSJ station, facilitating the semi-Lagrangian study of aerosol-cloud interactions.





Access to both stations is restricted, minimizing emissions from vehicular traffic. A map showing the location of both

stations is shown in Figure 1.

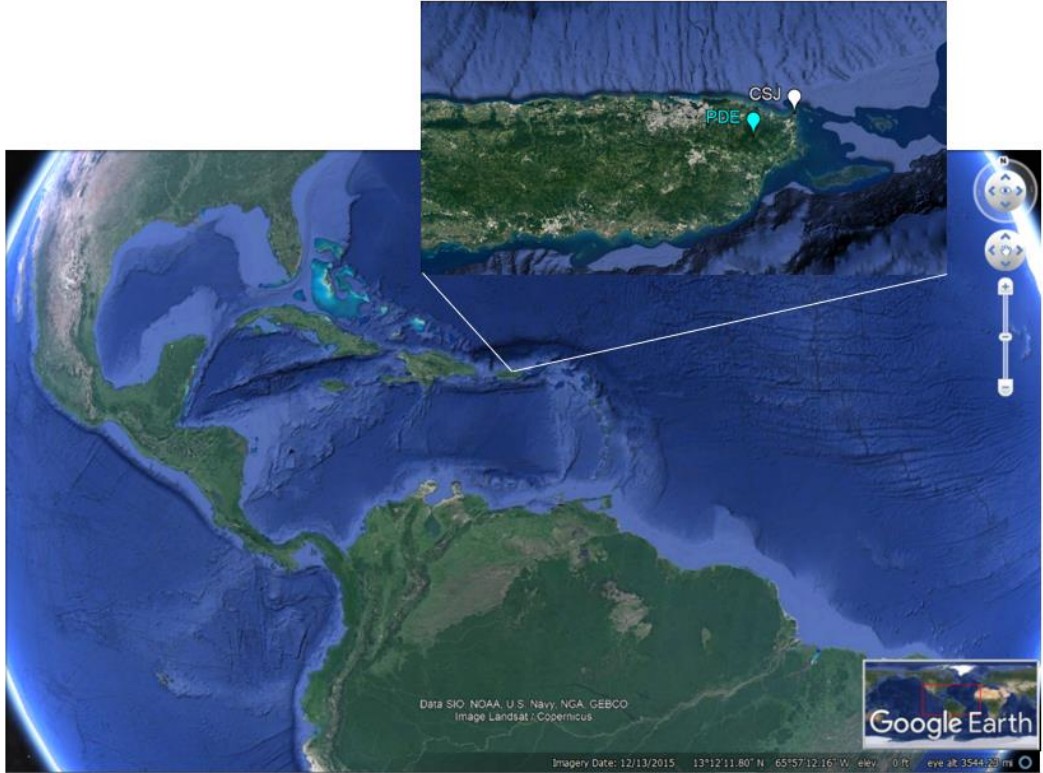

**Figure 1**: Map of the Caribbean basin with a zoom over Puerto Rico. The CSJ and PDE stations are highlighted in the map. This figure was generated using Google Earth Pro 7.3.

**2.2 Sampling Campaigns**

Sampling campaigns were held during the summer months, as these are the months when the island receives the influence of African dust. The sampling campaigns were: 2013 (June 09 to July 01), 2014 (August 20 to August 30), and 2015 (May 30 to August 29). The sampling campaign of 2015 extended to fall with the purpose of having more data with little to no dust influence.


At CSJ, the light scattering coefficients were measured with a nephelometer (TSI, model 3563) at three wavelengths (450, 550, and 700 nm) and the absorption coefficient with a Continuous Light Absorption Photometer (CLAP, NOAA), also at three similar wavelengths (467, 528, and 652 nm). The nephelometer and CLAP were positioned behind an impactor with aerodynamic size cuts of 1 and 10 μm (PM1 and PM10). The impactor switched

between sizes every six minutes. Since we are mainly interested in the coarse fraction of the aerosol, we used only PM10 data in our analysis.


At PDE, cloud water samples were collected using an aluminum Caltech active-strand cloud water collector (Al-CASCC2, Demoz et al., 1996). The sampler was exposed only during cloud events and rinsed thoroughly with Nanopure water before each collection period. Field blanks were also collected before each sample. Aliquots of the cloud water were stored in a freezer at -18ºC until analysis. Ion chromatography (Dionex ICS 1000 with conductivity detection) was used to determine the concentration of water-soluble ions. The ratio of sodium to calcium cations in sea salt was used to determine the amount of calcium that came from sea salt. The remaining calcium was attributed to non-sea salt sources (Wilson, 1975), mostly crustal.

Cloud microphysical properties were measured using a Backscatter Cloud Probe (BCP, DMT, Beswick et al., 2013) that measured over the size range of 5 – 90 µm. Sample volumes were corrected for wind speeds exceeding the tunnel velocity (Spiegel et al., 2014). The BCP measures the amount of light backscattered from individual cloud droplets illuminated with a laser. From the backscattered light intensity and counts, the droplet number, equivalent optical diameter (EOD), and liquid water content (LWC) were derived. The data were processed following the procedure detailed by Beswick et al., 2013.

Weather data were collected at both stations using Davis VantagePro2 Plus weather stations. Total rain, rain rate, wind direction, and wind speed were measured in 15-minute averaged intervals. The weather, aerosol and cloud microphysical data were averaged to hourly intervals for the analysis.

### 2.3 Classification of Sampling Periods

The aerosol light scattering coefficients were used as identifiers of aerosol particles whose optical properties differ from the background, marine aerosols. This approach assumes that dust particles will have scattering coefficients significantly higher than the average background. The average and standard deviation values for the 2013, 2014, and 2015 periods were calculated for the scattering coefficients at 550 nm as the midpoint of the measured range. Values that are more than one standard deviation above the average are classified as "high dust" cases, and those with one standard deviation or less than the average are labeled as "low dust" cases.

In addition, seven-day air mass back-trajectories were calculated using the Hybrid Single-Particle Lagrangian Integrated Trajectory (HYSPLIT) model (Draxler & Hess, 1998; Stein et al., 2015; Rolph et al., 2017) in order to identify the history of air masses prior to their arrival at the PDE site. GDAS reanalysis with a 0.5º resolution was used for the input meteorological data and the trajectories were run in single trajectory normal mode every two hours backwards 168 hours, with a 10,000 m agl top of model limit. The overall accuracy of the back-trajectories position depends on the wind fields taken from the meteorological model, interpolation of wind velocity from the grid point to the actual trajectory position (Rolph and Draxler, 1990) and truncation errors and has been estimated to be about 15-20% (Stohl et al., 2002).



The meteorological parameters that were derived along the trajectories were temperature, pressure, relative humidity, and rainfall. The air mass origins are classified according to where the air was located seven days before reaching the site.

The metrics selected for assessing the environmental effects of these air masses of differing origin were the scattering and absorption coefficients with derived Ångström exponents to characterize the aerosol properties, the droplet size distribution, and liquid water content to describe the cloud microphysical properties and the concentration of water-soluble ions to characterize the cloud water chemistry. The hourly latitude, longitude, altitude, and precipitation along the back trajectories are the metrics evaluated for characterizing the air mass properties.

### 2.4 Data Quality Assurance

The BCP data were recorded every second and reduced to 10-minute averages. Beswick et al. (2013) provide a detailed error analysis of the BCP measurements, in particular the uncertainty associated with deriving the size from the light scattering signal. For this study, the BCP was installed in a wind tunnel with constant airflow and orientation towards the prevailing wind direction. Laboratory tests of different wind speeds measured inside the tunnel showed that this implementation enhanced the outside wind speed by a factor of 1.4 inside the tunnel. The average corrected airspeed was $11.2 \pm 1.8$ m s$^{-1}$.

To ensure a sampling uncertainty of 10% or less, a population of 100 particles per sampling interval is needed. Coincidence errors become significant, $> 10\%$ undercounting, when droplet concentrations exceed 500 cm$^{-3}$ (Beswick et al., 2013); however, the PDE cloud concentrations rarely exceeded 500 cm$^{-3}$. The overall estimated uncertainties in number concentration, EOD and LWC, using error propagation, are 15%, 20%, and 40%, respectively.

At CSJ, nephelometer and CLAP data were also gathered at one-second intervals and reduced to six-minutes and hourly averages. The nephelometer data were corrected for the truncation error (Anderson and Ogren, 1998), and the CLAP data were corrected for enhanced absorption artifacts (Bond et al., 1999; Ogren, 2010). Data from both instruments were adjusted to standard temperature and pressure.

### 2.5 Statistical Analysis

The directly measured and derived parameters were evaluated to determine if their values were Gaussian distributed in order to select parametric or non-parametric statistical methods to use in testing the null hypothesis. This was done using the Shapiro-Wilk methodology (Shapiro and Wilk, 1965). Given that the majority of the metrics had frequency distributions that were not Gaussian, the Mann-Whitney, two-tailed test was used to determine if the average values of the selected parameters were statistically different with respect to the air mass origins.

### 3 Results



### 3.1 Aerosol Properties

Based upon the criteria described previously, using the light scattering coefficients at a wavelength of 550 nm, three periods of high dust (H1 - H3) and four of low dust (L1 - L4) were identified for 2013, one period of high dust (H4) in 2014 and twelve high dust periods (H5 – H16) and five low dust periods (L5 – L9) were identified in 2015. Figure 2a-c shows the time series for the scattering coefficients at three wavelengths (colored curves) and the Ångström scattering exponents derived from these coefficients (black). The orange boxes identify the periods of light scattering exceeding the selected threshold to designate dust. The blue boxes encompass low light scattering events. The solid arrows in Fig. 2c mark the days selected for a more in-depth analysis of the air mass origins on those days and the associated cloud properties, as will be discussed in later sections.

There are no regular trends in the optical properties that can be linked to daily variations in local emissions, indicating that the observed fluctuations are related to larger-scale changes in the air masses. The trends are irregular and marked by sharp increases and decreases, such as those seen in 2013 on days 165 (increase) and 167 (decrease). The durations of the events are also quite variable, several lasting more than 24 hours in 2013 and 2014, while in 2015 there are many events that are only several hours long. It is possible that some of the short events that are close to each other are actually related, i.e., coming from the same dust event but interrupted by brief shifts in the local circulation patterns before returning to the broader scale flows that are bringing the dust.



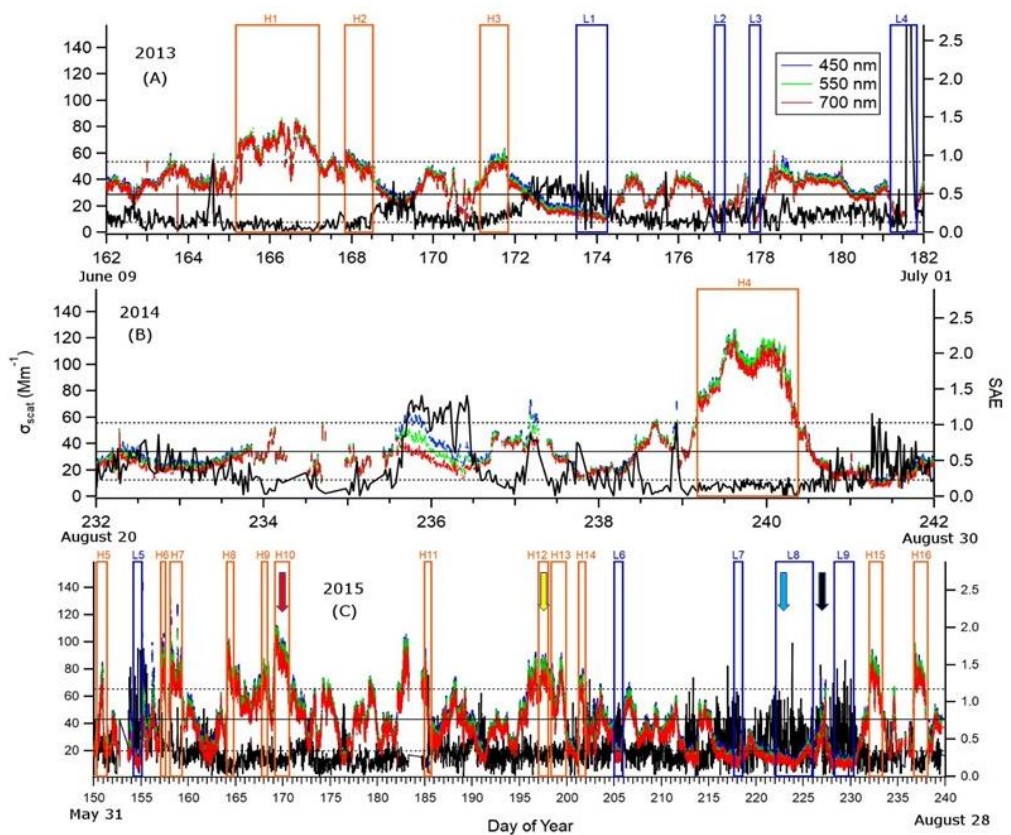


**Figure 2**. Aerosol scattering coefficients and SAE measured at CSJ during the summers of a) 2013, b)2014 and c)2015. The orange and blue squares identify the periods of high and low dust, respectively. The arrows in the 2015 time series identify the periods whose cloud properties were analyzed in detail when the air origins were from the African Sahel (red), Saharan Desert (orange), Atlantic Ocean (blue) and Southeast of Puerto Rico (black). These

periods correspond to those evaluated in the back trajectory analysis.

Figure 3 shows box plots for the parameters studied for high and low dust periods. Both the scattering and absorption coefficients were considerably higher for high dust periods. The SAE was lower for high dust periods, which indicates the presence of larger particles than in low dust periods. The AAE for high dust periods was usually

above two, while on low dust periods were usually below two. Both the low SAE and high AAE values confirm the presence of dust in high dust periods (Cazorla et al., 2013).

Non-sea salt calcium (nss-$Ca^{2+}$) in cloud water samples, collected in periods with high dust, was in considerably higher concentration than those taken in low dust periods (average of 83 vs. 0.97 μeq/L), validating the

presence of minerals in periods designated as high dust. A similar finding was previously reported in several other





studies (Gioda et al., 2013; Valle-Díaz et al., 2016). It is known that mineral dust contains large amounts of calcium (Scheuvens et al., 2013), so nss-Ca$^{2+}$ can be used as a proxy for mineral dust.

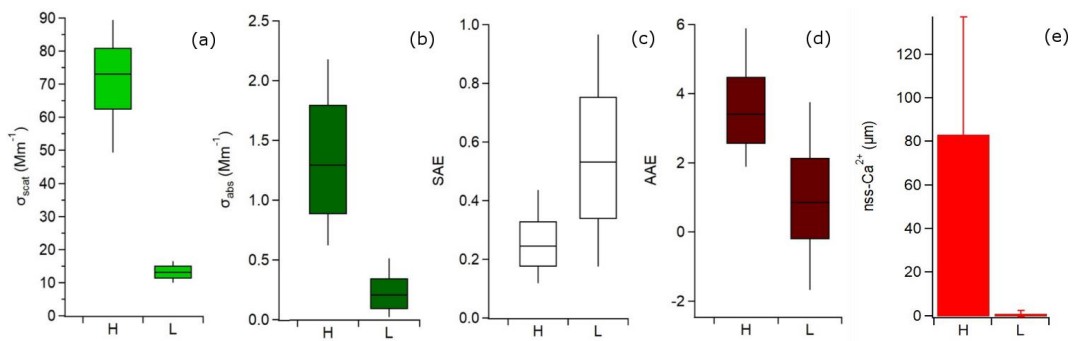


**Figure 3.** Box plots for periods identified as low and high dust for a) scattering and b) absorption coefficients, c) SAE, d) AAE and e) non-sea salt calcium concentration.

All the metric variables showed statistically significant differences between high and low dust periods (p < 0.001; α = 0.05) Table 1.1 shows the averages and standard deviations of the metric variables for high and low dust periods.

**Table 1.** Average and standard deviation for aerosol optical properties for the sampling periods of 2013, 2014, and 2015.

| Variable | Year | Average | Standard Deviation |
|---|---|---|---|
| Scattering 550 (Mm-1) | 2013 | 30.3 | 14.4 |
|  | 2014 | 34.2 | 20.3 |
|  | 2015 | 42.8 | 23.1 |
| Absorption 550 (Mm-1) | 2013 | 0.6 | 0.8 |
|  | 2014 | 2.0 | 2.8 |
|  | 2015 | 1.1 | 1.2 |
| SAE | 2013 | 0.5 | 0.4 |
|  | 2014 | 0.5 | 0.4 |
|  | 2015 | 0.5 | 0.4 |
| AAE | 2013 | 1.3 | 2.3 |
|  | 2014 | 1.2 | 2.3 |



| | 2015 | 2.6 | 2.2 |
|---|---|---|---|

### 3.2 Air Mass Origin Analysis

An analysis of back trajectories from the PDE site was used to track the histories of air masses that were arriving during periods of cloud at the site. Back trajectories were computed every two hours. Those time periods when six or more consecutive trajectories of air had been over the same general region, seven days previously, are used to identify air whose aerosol particles were in all likelihood distinctly different in physical and chemical properties. The four regions that were identified in this manner are: 1) Atlantic Ocean (AO), 2) Southeast (SE), 3)

Saharan Desert (SD) and 4) African Sahel (AS). The days that were selected were those during which there was already a cloud present at the PDE.

     Figures 4a-c show the averages of the four air mass types as seven-day, horizontal back trajectories, vertical locations of the air along the trajectories and accumulated precipitation, respectively. The numbers in parentheses after

the air mass name in the legend correspond to the number of consecutive two-hour trajectories averaged. Figure 4a highlights the primary difference between the AO and SE airmasses. They both are over the ocean seven days prior, in the mid-latitudes and more than 1500 km from the nearest land mass; however, the AO air arrives at the PDE from the northeast whereas the SE air first travels south of Puerto Rico's latitude before turning back to the northwest. This latter trajectory brings it over the inhabited islands to the east and south of Puerto Rico.


     Although the SD and AS trajectories are over the African continent seven days prior to arriving at PDE, the SD air masses were in the region over the desert while the AS air is more tropical and over the region known as the African Sahel. The other metric that clearly distinguishes the air masses from one another is the altitude history that is documented in Fig. 4b. Of the four types, the SE air masses are the only ones that have traveled much of their seven-

day trajectory near the surface, especially the three days before reaching the island. Although their histories vary, as illustrated by the vertical bars that mark the standard deviations about the mean, the AO, SD and AS air masses are all descending as they reach PDE. This is an important observation due to the different forcing mechanisms that lead to cloud formation, a factor that will be discussed below.

The altitude histories are also important indicators of the possible sources of aerosols that arrive at PDE and may influence the cloud properties. For example, SE air masses spent more than three days of their travel below 1000 m and would presumably be influenced by mixtures of sea spray and anthropogenic emissions as they passed over the inhabited islands east and southeast of Puerto Rico. On the other hand, neither the AO or AS trajectories took them close to the surface, suggesting that the aerosols in these air masses were either introduced from the surface at some

point even earlier in the air mass history than seven days or, alternatively, convective activity could transport aerosols vertically from the surface at various points along the travel. As indicated by the very large standard deviations on the



SD airmass trajectory, many of those air masses were near the surface seven days previously, so that desert dust lofted from the surface would be a major source of the aerosols that arrived at PDE.

The amount of precipitation that was derived with the HYSPLIT model is shown for the four air mass types in Fig. 4c, represented here as total accumulated rainfall along the track. The trends in accumulated precipitation reveal a number of interesting details about meteorological events along the trajectories: 1) the majority of AO, SD, and AS precipitation events occurred more than five days before arriving at PDE, 2) there was little precipitation along the SE air mass until 30 hours before arriving at PDE, 3) the AS air masses experienced the most precipitation, the majority
of which occurred within a day after leaving the African continent and 4) the AO and SD air masses were generally very dry, i.e. very little precipitation events transpired over the seven-day journey. The significance of these precipitation events on the subsequent impact of these air mass is discussed below.

    One point to mention is that there were no volcanic eruptions or ash plumes reported for the Soufriere Hills
volcano on the island of Montserrat, upwind of Puerto Rico, from the Montserrat Volcanic Observatory (www.mvo.ms) during the three years of measurements. Given that the SE type airmasses passed over this region, the lack of volcanic aerosols is important to note since these would have been an additional source of aerosols that might have impacted cloud microphysical properties at PDE.


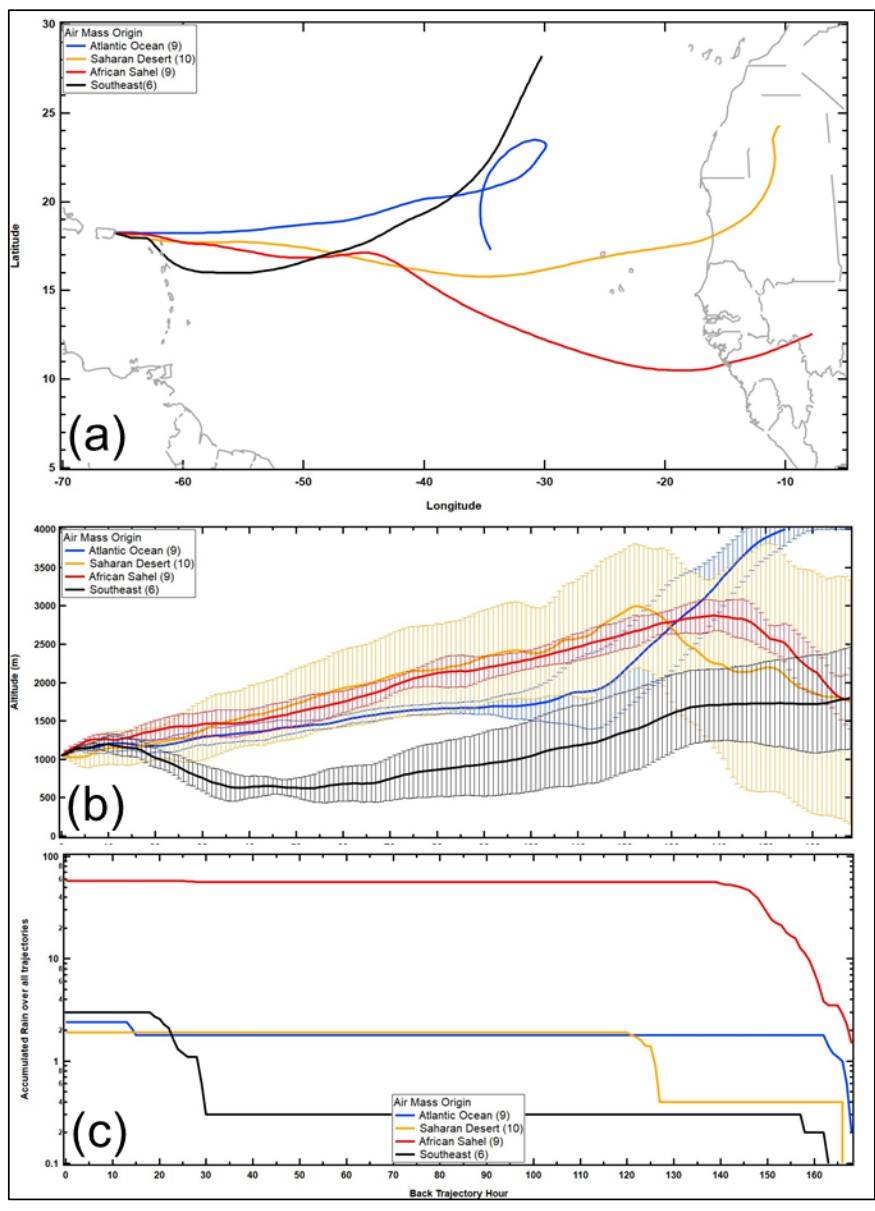

**Figure 4.** HYSPLIT back-trajectory analysis. a) The 7-day averaged back trajectories are shown for air masses with different origins: AO (blue curve), SD (yellow), AS (red) and approaching PDE from the Southeast (black). The numbers in parentheses indicate the number of sequential averaged for each type. b) The average altitudes during the 7-day travel before reaching PDE are shown as a function of back trajectory hour for the four air mass types. The vertical bars are one standard deviation about the mean value. c) The accumulated rain that has fallen out of the air mass, as simulated by the HYSPLIT model, is shown as a function of hour before reaching PDE.




### 3.3 Cloud Analysis

The hypothesis that drives the analysis in this study is that the microphysical properties of the mountaintop clouds in Puerto Rico are not strongly dependent on the air mass histories. In order to test this hypothesis, we have taken a case study approach whereby four periods were selected during which clouds were already present on the mountaintop when the air with different histories arrived. The cloud properties are evaluated as this new air arrives over a period lasting anywhere from 12 to 20 hours, the time periods when the back trajectory analysis has shown that

the air is arriving from the same source.

        An alternative, and perhaps preferable, analysis method would be to identify periods when clouds were formed under conditions related to the four air mass types and then evaluate the subsequent evolution; however, such conditions could not be unambiguously identified for all four types with the current data set. Hence, the analysis

evaluates changes in the size distributions relative to the cloud properties that existed at the time when the new air mass arrived. This is analogous to what is done during cloud seeding programs, where the material is introduced to an existing cloud in order to alter the subsequent evolution of cloud properties.

        Figures 5 and 6 highlights how the size distributions of number concentration, $N_c$ (Fig. 5), and liquid water

content, LWC (Fig. 6), evolve with the introduction of new air from the four different origins. The time series are drawn starting four hours before the new air masses entered the cloud and continue for 18 more hours. The number and liquid water concentrations are color-coded as $cm^{-3}$ and $mg\ m^{-3}$, respectively. The hourly average $N_c$ and MVD are also drawn on Figs. 5 and 6, respectively (solid black curves). The dashed red lines on the figures demarcate the period of time that the air is arriving from the designated air mass origin.




**Figure 5.** Time series of the size distribution of number concentrations for air masses with origins of a) AO, b) SD, c) AS and d) SE. The color coding designates the value of the concentration. The average number concentration is

drawn as the solid black curve and the red dashed curve shows the period when the new air mass was arriving.



**Figure 6.** The same as Figure 5 but for the size distribution of liquid water content (LWC). The color coding designates the value of the LWC (mg m⁻³). The average median volume diameter (MVD) is drawn as the solid black curve.


Each of the events occurs at somewhat different times of day. The AO, AD, AS and SE air is introduced to the clouds at 1400, 0200, 2000 and 1000 UTC (1000, 2200, 1600 and 0600 Local time), respectively. Previous studies



have documented that clouds can form at any time of the day and last for many hours or even days (Allan et al., 2008). Note, all times reported are in UTC.


There are changes in the cloud properties after the new air is introduced in all four cases; however, the magnitude and trends differ. Clear differences are seen in the shapes of the size distributions over time following the introduction of air with new time histories. The AO air mass arrives at 1400 Local Standard Time (LST) as a cloud is forming, and the $N_c$ increases rapidly during the first three to four hours, remains more or less constant over the next

10 hours, and then begins fluctuating by ±40 cm$^{-3}$ over the last six hours that the air mass is coming from the AO. The average number concentration and MVD during the 18 hours of AO were 80 cm$^{-3}$ and 11 µm, respectively.

The SD air arrived at two o'clock in the morning, LST, to an already formed cloud and continued until ten o'clock in the evening. At the time this air arrived, the cloud $N_c$ and MVD had average values of 80 cm$^{-3}$ and 10 µm,

respectively. The $N_c$ decreased to 60 cm$^{-3}$ as the new air arrived and stayed at that value for the next eight hours before becoming highly variable during the remaining 12 hours. The MVD stayed constant for six hours after the air mass arrival but then began decreasing throughout the remainder of the air masses lifetime at PDE until reaching a minimum of 5 µm.

The cloud that was already formed at PDE displayed no changes in the microphysical properties when the air mass with the AS history arrived at 2000 LST, and these properties remain almost constant over the next three hours with an average $N_c$ of 60 cm$^{-3}$ and MVD of 10 µm. At that time, the $N_c$ decreased to 40 cm$^{-3}$, but the MVD remained constant. The cloud persisted another 12 hours with very little change in $N_c$, LWC or MVD at which time it dissipated while the AS air was still present for another two hours.


The SE air mass arrived at 1000 LST during a period of sparse cloud that remains very low in concentration with average $N_c < 10$ cm$^{-3}$ for five hours, at which time the number concentration began to increase, reaching a maximum of 160 cm$^{-3}$ at the end of the air mass's time over the PDE. The cloud then rapidly dissipated in the hours following the end of this air mass of SE history. Of note also is that over the six-hour period of increasing

concentration, the average MVD also increased by a factor of three, from 5 to 15 µm.

## 4 Discussion

The times series of the microphysical properties of clouds that evolved in air with different air mass histories show that the size distributions of number and mass concentrations, their associated average $N_c$ and MVD and lifetimes

are quite different. Clouds are formed on CCN as air masses cool and reach supersaturated conditions; hence, their characteristics will depend on the thermodynamics of the air as well as the physical and chemical properties of the aerosols carried by the air parcels.



The trends in the bulk properties derived from the size distributions, i.e., the $N_c$, LWC and MVD, are metrics that can clarify differences in the behavior of clouds under the influence of different air mass histories, as illustrated in Fig. 7. In this figure, the deviations (anomalies) from the average values are shown as a function of time, where the averages are calculated for the period that the air mass is arriving at PDE. The plotted time series begins four hours prior to the air mass arrival and extend an additional 20 hours. Starting and ending values of -100% demarcate the time periods of the air masses with individual durations.

Due to the arrival of the AO air at PDE at the same time as the cloud was forming, the concentration deviation (Fig. 7a), $\Delta N_c$, from the average is negative, increasing to an approximately +25% deviation that lasts eight hours before once again decreasing below the average. The $\Delta$LWC (Fig. 7b) follows a similar trend, correlating closely to the $\Delta N_c$; however, the maximum $\Delta$LWC is +75% compared to +30% for the $\Delta N_c$. This difference in the maximum deviations is a result of changes in the $\Delta$MVD (Fig. 7c), whose trend follows those of the $\Delta N_c$ and $\Delta N_c$, but reaches a maximum deviation of 20%. Since LWC ~ $(N_c)(MVD)^3$, the $\Delta$LWC will experience a much larger change with deviations in MVD than in $N_c$ and by using the same calculation used in the propagation of errors, i.e., the root summed square (RSS), $\Delta$LWC ~ $\Delta N_c + 3\ \Delta$MVD = 80%.

The AD air masses arrived at PDE and encountered forming cloud whose concentration is 50% higher than the average during the 20-hour residence of the new air, but the $\Delta N_c$ gradually decreases for eight hours, at which time it becomes more negative, reaching a minimum of -90%, 12 hours after this air mass reached the mountain top. During the next six hours, the deviation trends back towards the average and is nearly zero at the end of this air masses residence at PDE. Unlike the trends in the AO-impacted clouds, not only do the $\Delta N_c$ and $\Delta$LWC trends track one another, they vary by approximately the same percentage during the first five hours while the $\Delta$MVD remains constant at 25%. This indicates that the changes in LWC are almost completely a result of variations in $N_c$ the first five hours, after which the much sharper decrease in the $\Delta$LWC (Fig. 7b) is due to the same rapid decrease in the $\Delta$MVD (Fig. 7c).

When the air of AS history reaches the mountaintop that is already in cloud, the $N_c$, LWC, and MVDs are 75%, 125%, and 25% higher, respectively, than their averaged values over the period and maintain these values for a couple of hours before decreasing. The $\Delta N_c$ and $\Delta$LWC decrease to a minimum of -35% over eight hours, while the $\Delta$MVD decreases rapidly to 0% in less than an hour and remains constant for eight hours. During the remaining time that the AS air is reaching PDE the $\Delta N_c$ increases from negative to positive, but even though the $\Delta$LWC trend is positive, the $\Delta$MVD is not so that the LWC never again exceeds the average. This underscores the importance of the droplet size in controlling the LWC.





**Figure 7.** The deviations in a) number concentration, b) LWC and c) MVD from their average values over the time
period of air mass intrusion are summarized in these figures for air mass types, in percent change.

The trends in the $\Delta N_c$, $\Delta LWC$, and the $\Delta MVD$ when the clouds are under the influence of the air masses with
the SE history are the opposite of those experienced under AD and AS air masses. The initial deviations in all three



of these parameters -100% from the average because the initial cloud concentration that the air encountered was very
low. The initial increase in the $\Delta$MVD is not reflected in the $\Delta N_c$ or $\Delta$LWC because of the low concentrations. Once
the cloud begins developing, however, the $\Delta N_c$ and $\Delta$LWC continue increasing, along with the $\Delta$MVD until the end
of the SE residency.

To encapsulate in a few sentences the physics behind the trends that we see in the deviations of the cloud
microphysical parameters: 1) when $\Delta N_c$ increases, new droplets are activating, 2) when $\Delta$LWC is increasing it can be
due to increases in $\Delta N_c$ and $\Delta$MVD (droplets growing in size by condensation or coalescence), 3) when $\Delta N_c$ decreases,
droplets are being removed by evaporation or precipitation and 4) when $\Delta$LWC is decreasing, it may be a result of
decreasing $N_c$ and MVD.

In this context, the impact of the air masses with four different histories is:

- AO – Activation and growth of new droplets followed by eventual evaporation due to entrainment and mixing
  of subsaturated air;
- SD – Initial entrainment of subsaturated air that led to removal of droplets by evaporation then new activation
  of CCN, either the residuals of the evaporated droplets or those residuals in addition to CCN introduced by
  the SD air mass, then further entrainment of subsaturated air that led to droplet evaporation and the dispersion
  of the cloud;
- AS – The impact of this air mass is largely to remove the existing droplet by evaporation with only a brief
  period when it appears that some new activation occurred;
- SE – The arrival of the new air eventually resulted in the activation of droplets in much higher concentrations
  than seen in the clouds under the influence of the other air masses and these CCN are being continuously
  activated throughout the period, suggesting that the supersaturation is continuing to increase and smaller
  sized CCN are being activated. At the same time, the droplets continue to grow, as evidenced by the
  increasing MVD throughout the residency of this air at PDE.

Following this summary, the trends in the microphysical properties suggest that only the AO and SE air masses
are introducing new CCN whereas neither of the air masses of African history have CCN in sufficient concentration
or difference in properties to generate new cloud droplets.

Previous studies at this location looking at cloud properties influenced by different air masses and aerosol types
have come to mixed results. Eugster et al., 2006 and Spiegel et al., 2014 arrived at the conclusion that the presence of
African dust particles could be acting as cloud condensation nuclei and thus being the main responsible for different
cloud microphysical properties. However, our results show otherwise and are in line with those seen in Raga et al.,
2016, which results show that air mass history, mainly precipitation and time below 500 m prior to reaching the
sampling site, had a better relationship with the number of CCN.



As previously mentioned, not only are the properties of the aerosols important when assessing the effect of the air masses, but the dynamics and thermodynamics of these air masses are also potentially of relevance. The Puerto Rico mountaintop clouds are formed from rising and cooling air brought about by surface heating that leads to convection, orographic lifting or by cold frontal passage. Only the first two mechanisms occurred during the summers

of 2013-2015. Given that surface heating is a daytime phenomenon, the clouds that formed during the incursions of air masses with AO and SE histories, around 1600 LST on both of these occasions, are likely the result of convective activity. In addition, in both these cases, the clouds persisted well into the evening, even after these particular air masses were no longer in residence. Hence, the clouds that were already on the mountain during the SD and AS incursions were also most probably those that had formed earlier in the day by convective activities. So before

finalizing our conclusions about how the clouds in the four cases were altered by the changing air mass histories, it is important to evaluate if there were obvious differences in the winds or thermodynamic variables.

Figure 8 summarizes the wind speeds and directions prior to and during the incursions of air masses, where the vertical dashed lines separate the time periods before and after the new air arrived. Other than the average wind speed

being somewhat lower when the air is from the African Sahel, there are no obvious shifts in the velocity that can be associated with differences in the cloud properties. A significant increase in wind speed might lead to more turbulence and entrainment, or a large change in direction could be associated with a different magnitude of forcing; however, since neither of these shifts occurred, it suggests that atmospheric dynamics is not playing a significant role in the cloud evolution in these cases.

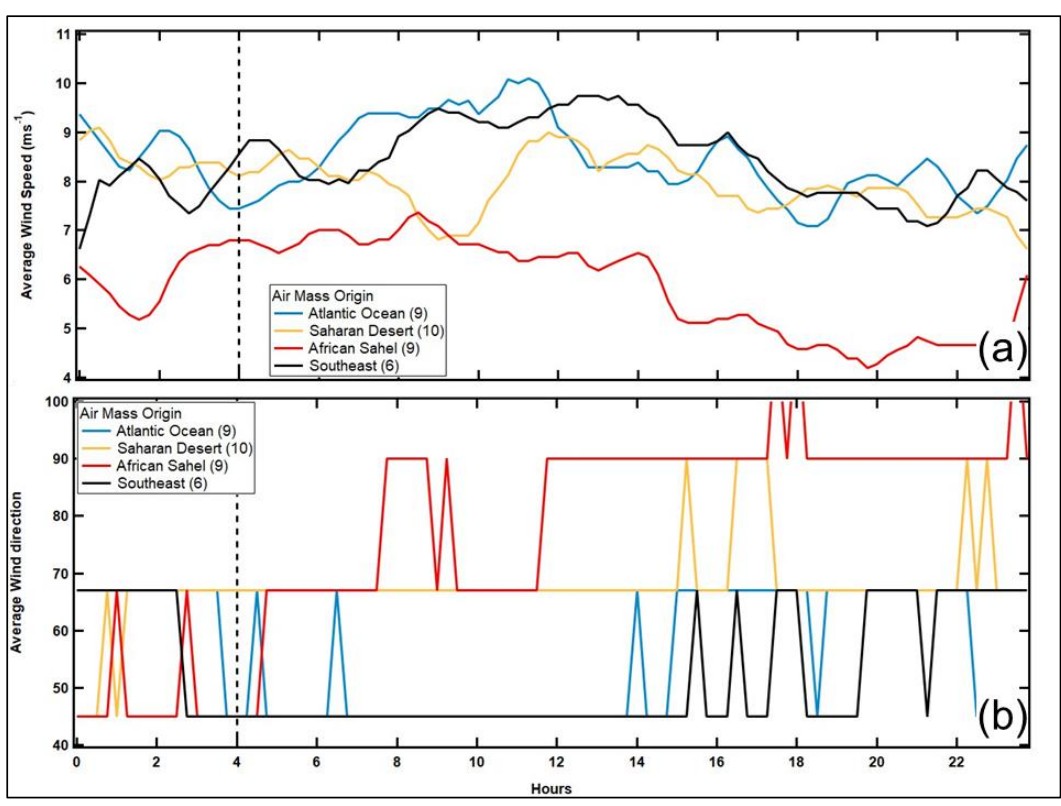

**Figure 8.** The average a) wind speed and b) wind direction during the time period of air mass intrusions are summarized in these figures for the air mass types.

The thermodynamic history of the air masses, i.e., the temperature and relative humidity (RH), tells a different story as is illustrated in Fig. 9 where the back-trajectory analysis summarizes how these two parameters evolved in time prior to the arrival of air at PDE.

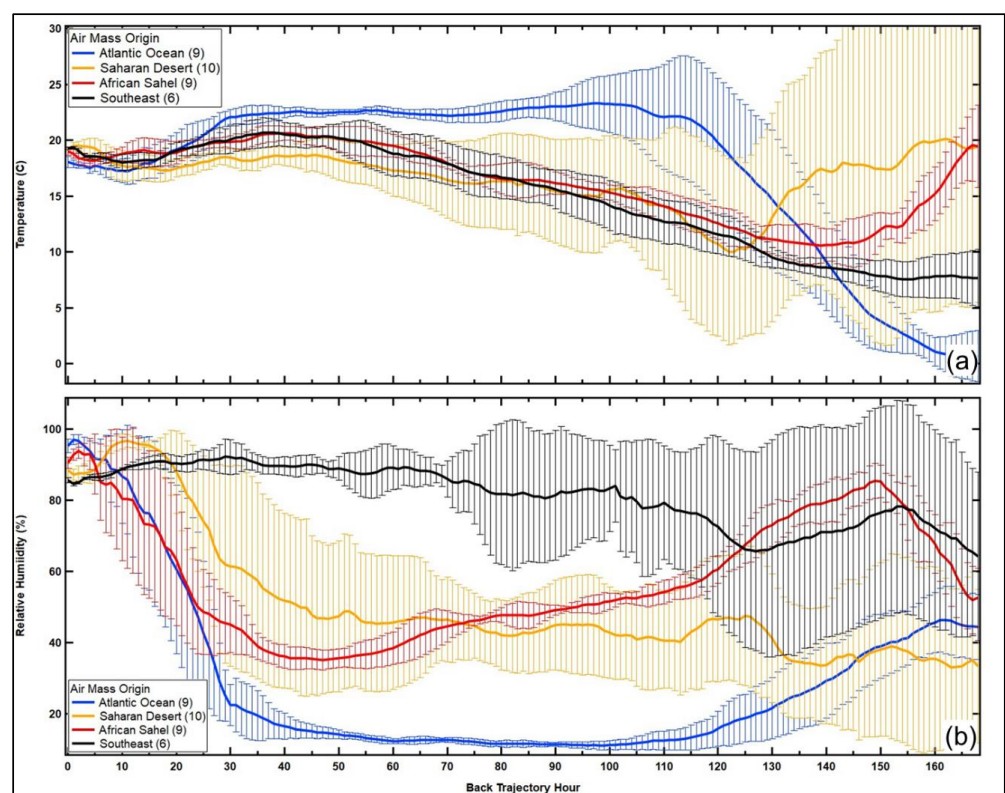

**Figure 9.** Seven-day, HYSPLIT back trajectory analysis of average a) temperature and b) relative humidity. The vertical bars are standard deviations around the averages. The numbers in parentheses after the trajectory labels are the number of consecutive trajectories that were averaged for each air mass type.

As expected, the temperature histories follow those of altitude that were drawn in Fig. 4b, i.e., the African air masses, SD and AS, had been at lower altitudes and warmer temperatures seven days prior to reaching while the AO and SE over the Atlantic were much higher and colder. All of the trajectories converge on an average temperature of 18° ± 2°, 24 hours prior to arrival at PDE so that there are no differences that might impact the subsequent temperature of the cloud when mixed with this air.

The RH histories of the air masses are distinctly different from one another (Fig. 9b) and reflect to some degree both their temperature and altitude trends, in particular the air mass with the SE history that was within 1000 m of the ocean surface throughout most of its seven-day travel. The proximity to this water vapor source kept its RH above 80% from three days back until reaching PDE. The decrease in RH at 20 hours is a result of its ascent as it approached the island.

The RH trends of the AO, SD and AS air masses are more puzzling and would require an analysis beyond the scope of the current study to explain the rapid increase in the RH of these air mass beginning at 30 hours that leads





to RH values above 90% when these parcels reach the measurement site. If the RH had been much lower than 90% when arriving at PDE, mixing it into the existing cloud would have led to more rapid evaporation of the droplets and dissipation of the cloud. However, as the time series of the size distributions shows, the dissipation happened hours after the arrival of the air masses, so mixing will decrease the concentration, not due to removal of droplets by evaporation, but just by mixing in droplet free air.

It is important to acknowledge that we have imposed conditions on the analysis that might have biased the results towards the higher RH air at the time of arrival or one hour before; however, as the time traces show in Fig. 9b, there was already a strong, positive trend of humidification many hours prior to the air mass arrivals.

To conclude the discussion, we return to the analysis of the accumulation of precipitation along the air mass trajectories that was first introduced in Fig. 4c. The rainfall that is reported from the HYSPLIT model is an estimate based on the humidity fields and atmospheric cooling derived from the meteorological fields that are used when the model is run. Although these are not observed values, previously published comparisons between in situ and satellite measurements have shown that it is a reasonable approximation that can be used to evaluate precipitation along the trajectories. In the present study, we are interested in how precipitation affects the population of CCN in the air mass that will eventually reach Puerto Rico. The model predicts that the majority of rainfall is happening in the SD and AS air masses more than five days prior to their arrival at PDE. This suggests that some fraction of the dust and biomass burning aerosols that might have been picked up over Africa will be removed by precipitation, well before they will have a chance to influence the mountaintop clouds in Puerto Rico. On the other hand, if this precipitation is a result of deep convection, while rainfall might remove some CCN, the vertical air motions that lead to the formation of clouds and precipitations will also transport CCN that are not washed out. This latter mechanism of aerosol transport is particularly important for those air masses with AO and AS histories whose altitudes from seven days forward never brought them near the surface as was seen with the SD and SE air masses.

The evidence that argues for the insertion of new aerosols into the air masses by deep convection lies in the RH histories because the convection that would bring particles from the boundary layer into the free troposphere would also inject additional water vapor. Indeed, we see that six days back in the AS history, at the same time that the accumulated rain increases, the RH also increases, so that both water vapor and particles are being lifted vertically at the same time some rainfall is removing particles. On the other hand, the increase in rainfall accumulation experience by the AO and SD air masses is not accompanied by substantial RH increases.

Thus, to revise our former observations about the impact of these four types of air masses with different histories, the clouds that are impacted by AO air masses experienced an increase in concentration that can be associated with new CCN activation; however, given the evidence that these air masses had not been near the ocean surface, or received additional particles over the previous seven days, it is likely the increase in RH of this air mass as it approached Puerto Rico that provided additional moisture to the existing cloud. This fresh influx of water vapor conceivably could reactivate CCN residuals leftover from evaporating droplets. The evaluation of winds, temperature, and humidity do not change our original conclusion about the SE air masses that are bringing anthropogenic CCN



from the boundary layer that subsequently led to higher concentrations of droplets and that the primary influence of the SD and AS air is to dilute and eventually dissipate the clouds with no new droplets activated.

.

## 5 Summary and Conclusions

In an effort to expand the current database of measurements related to the interaction of aerosol particles

transported from the African continent to the Caribbean, and their subsequent impact on cloud formation and evolution, we conducted measurement campaigns on the Caribbean island of Puerto Rico during the summers of 2013, 2014, and 2015. On frequent occasions, the island was clearly inundated with air masses from the Saharan desert and African Sahel. Measurements were made at a coastal location of the aerosol optical properties and during these same periods, cloud microphysical properties were measured at a mountain site that was downwind from the coastal site.

The wind velocity, temperature, RH, and rain rate were measured at both sites. Based on the averaged wind directions, neither of these sites were influenced by anthropogenic emissions from local sources.

The aerosol optical properties and particle chemical composition measured at the coastal site were used to identify sixteen high dust and nine low dust events of African dust. During high dust periods, the scattering and absorption coefficients, AAE, and nss-$Ca^{2+}$ ion concentrations were higher, and the SAE were lower, than for low

dust periods.

The HYSPLIT back trajectory model was used to identify air masses with four distinctly different characteristics based on their geographic locations prior to arriving at the Pico de Este (PDE) measurement site: mid-

latitude Atlantic Ocean (AO), Sahara Desert (SD), African Sahel (AS) and arrival from southeast of Puerto Rico (SE). Time periods were selected for analysis during which these air masses were arriving continuously over durations ranging from 12-20 hours when there was a cloud present on the mountain peak. The cloud droplet size distributions were measured with a droplet spectrometer over the size range from 5-90 µm, from which the microphysical properties, total number concentration, $N_c$, liquid water content, LWC, and median volume diameters, MVD, were

derived as a function of time.

Based on the atmospheric dynamics and thermodynamics, and the trends in the microphysical properties, we arrive at the following conclusions regarding the impact of these air masses with four different histories on cloud evolution:

- The AO air masses influence the reactivation of existing CCN as a result of the mixing of high humidity air with the resident cloud and not due to the introduction of new CCN.
- The initial entrainment of SD air parcels leads to removal of droplets by evaporation then new activation of CCN, either the residuals of the evaporated droplets or those residuals in addition to CCN introduced by the SD air mass, then further entrainment of subsaturated air that leads to droplet evaporation and the dispersion

of the cloud.



- The impact of air masses with AS history is largely to remove the existing droplets by evaporation with only a brief period when it appears that some new droplet activation occurred, most likely on CCN that were residuals of evaporated droplets.

- The arrival of the new air along the SE trajectories eventually resulted in the activation of droplets in much higher concentrations than seen in the clouds under the influence of the other air masses and these CCN are being continuously activated throughout the period, suggesting that the supersaturation is continuing to increase and smaller sized CCN are being activated. At the same time, the droplets continue to grow, as evidenced by the increasing MVD throughout the residency of this air at PDE.

These conclusions do not support those by Spiegel et al. (2014) who concluded that microphysical properties of clouds at PDE were significantly altered by African dust. They do support the arguments put forward by Raga et al. (2016) who used similar measurements at PDE from the summer of 2011 to conclude that the meteorology has a larger impact on the cloud microphysics than the properties of the CCN.

Clearly, the conclusions posted here are somewhat speculative in nature and require a much more detailed and long-term measurement program, coupled with cloud and chemistry models to validate these speculations. Nevertheless, the results are consistent with previous studies and, most importantly, provide a well-documented set of measurements to enhance the current data set of similar observations in the Caribbean.

**6 Data Availability**

Data available upon request.

**7 Author Contribution**

ETD, DB, and OLMB designed the field sampling campaign. OLMB designed and acquired the funding for
the project. ETD, DB, and OLMB performed the data analysis and interpretation of results. ETD prepared the manuscript with contributions from all authors.

**8 Competing Interest**

Darrel Baumgardner is one of the founders of Droplet Measurement Technologies.


**9 Disclaimer**

Any mention of a company or product does not constitute an endorsement form the authors.

**10 Acknowledgments**

The authors acknowledge the Luquillo Critical Zone Observatory (LCZO; NSF EAR Grant 1331841) for the financial support for the project, as well as the Long-Term Ecological Research project, Bridge to the Doctorate



Program (Grant HRD1139888), and the NASA Puerto Rico Space Grant Fellowship and Scholarship Program (Grant NNX15AI11H) for the student support.

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
