# Peer review of "Measurement Report: Impact of African Aerosol Particles on Cloud Evolution in a Tropical Montane Cloud Forest in the Caribbean"

_Atmospheric Chemistry and Physics, 2021_

## Author Response (AR1)

Torres-Delgado et al. "Measurement Report: Impact of African Aerosol Particles on Cloud Evolution in a Tropical Montane Cloud Forest in the Caribbean"

This study investigates the impact of African aerosol particles on cloud evolution in a tropical montane cloud forest in the Caribbean based on case studies using measurements from Aircraft, satellite and back-trajectory model. It shows interesting results while I do not think the conclusion are very robust at this moment. Personally, I think this study is worthy to be published in ACP after a major revision.

Introduction part: The authors may consider reading and citing the recent studies in addition to the studies at early years. For example, the direct solar radiation effect of dust from Africa on the meteorology (Sun et al., 2020, doi: 10.1029/2020JD033454).

The authors understand that the suggested reference is relevant for the article, and it has been cited now in the introduction.

Line 23-25, The authors mentioned "Great studies" but only providing one reference. More references are necessary here. Actually, there are huge amount of studies regarding aerosol-cloud-radiation interaction studies, along with the uncertainties, such as Stephens (2005, doi:10.1175/JCLI-3243.1), Garrett and Zhao (2006, doi:10.1038/nature04636), Li et al. (2011, doi:10.1038/ngeo1313), and Li et al. (2019, doi:10.1029/2019JD030758).

The authors would like to respectfully correct the reviewer, as the text reads "great strides". However, the suggested articles were considered, and some were cited as they were relevant to the study.

Line 38-39, could the authors provide the exact or estimated hours for the cloud existence? That would be helpful to the readers.

Previous studies have documented the frequency of clouds and their duration during various seasons. These are now referenced in the text.

Line 54, "cloud properties"

Corrected

Section 2.1 About the sampling site map: If possible, the trade winds and the metropolitan area could be indicated in the map.

The metropolitan area (San Juan) has been highlighted and a compass has been included in the map. Trade winds have not been included as the text states that trade winds come from the northeast (Section 2.1 – Sampling Sites).

Line 86-91, I wonder if PM10-PM1 is better to represent the dust aerosols rather than PM10. If it is, the authors may consider using the difference.

The main pollution sources that may contribute to the PM1 are the nearby islands and Puerto Rico's metropolitan area. The contribution from the nearby islands is small and depends on the wind direction (SE) and the metropolitan area is downwind from the sampling station under most meteorological conditions except for the rare cold front. Since the winds mainly come from the E-NE directions, we do not consider anthropogenic pollution to be a major source of aerosols for these sites on the sampling days presented here. Also, the bulk of the dust aerosol is skewed towards coarse particles. Thus, we do not expect a big difference between PM10 or PM10-PM1.

Line 101-102, are the size range for diameters or radius?

Size range is for diameter. This information has been included in the methodology section.

Line 101-106, For the cloud microphysical properties measured by BCP, have the authors removed the observations within the first 2 bins as most methods used due to their large uncertainties?

The observations within the first two bins were not removed. However, in our dataset these bins sizes only account for a small fraction of the size distributions, as can be seen on figures 5 and 6.

Line 108-110, more information about the weather data is necessary and helpful.

We have change this section to read:

"Meteorological data were collected at both stations using Davis VantagePro2 Plus weather stations. Total rain, rain rate, wind direction, and wind speed were measured in 15-minute averaged intervals. The meteorology, aerosol and cloud microphysical data were averaged to hourly intervals for the analysis."

The meteorological stations are called "Weather Stations" by the manufacturer, but this is somewhat misleading since they really only measure local state variables and winds, not "Weather". Further in the manuscript we clearly describe general airflow conditions that are impacting the cloud formation and development, but since weather is a more general term that includes a variety of meteorological conditions, we do not try to describe weather patterns as that would be a major expansion of the text and not relevant to the focus of the study.

Line 116-118, I do not understand what do you mean "those with one standard deviation or less than the average are labeled as "low dust" cases". Please explain or modify.

A sentence that better explains this procedure has been included.

Line 120-127, please provide more information about the model setup (such as physics and dynamics modes, the released particle numbers, and so on), and whether you used the clustering. The errors could vary with the location, topography, and meteorology. As indicated by Zhao et al. (2009, doi:10.1029/2008JD011671) for California region, the trajectory uncertainty could be more than 20% just due to errors in meteorology (such as boundary layer height and winds).

The information requested by the reviewer does not apply to this study. We only used the trajectory analysis and did not use any particle modeling. The article by Zhao et al. 2009 was focused on transport of greenhouse gases, and the model they used (STILT) is optimized for dispersion of inert gases. On the other hand, the HYSPLIT model is optimized for calculating air mass trajectories. While a STILT-like configuration can be achieved using HYSPLIT, this is beyond the scope of this study. Clustering was not used.

Line 176-182, these are interesting. I wonder how large the air speeds are. Personally, I think local emissions could play more important role when the wind speeds are low.

For the sampling period, wind speed ranged from 2.20 to 12.10 m/s with an average of 7.24 m/s. We disagree with the reviewer's comment as through several years of studies on this site we have not found that local emissions are an important source. We have found that wind direction is more important in this regard, as long-range transported pollution can reach the site if wind directions come from the SE (nearby islands) or if we have cold fronts (common during winter, but rare during summer) and none were identified for the sampling periods presented here.

Line 192-197, SAE and AAE have not been defined yet. Please define them. Also, regarding the derivation of dust aerosols from low SAE and high AAE, recent studies could be referred, Yang et al. (2021a, b; doi:10.5194/acp-2020-921; doi:10.5194/acp-2020-1139), and Zheng et al. (2017, doi:10.5194/acp-17-13473-2017).

Terms have been defined. The suggested studies have been evaluated but they use the extinction Ångström exponent instead of the scattering and absorption Ångström exponents. In this study we used both values to help identify dust events as described by Cazorla et al., 2013, which is a different approach to that used in the suggested articles.

Figure 4. the figure quality need improve.

Since the reviewer has not been specific about what needs improving, we have not changed the figure but did make a small correction to the figure description for clarity.

Line 286-299, For aerosol-cloud interaction study, or for cloud seeding studies, it is necessary for us to know the cloud status including the cloud phase (warm clouds or super-cooled liquid/mixed-phase clouds) and cloud microphysical properties. As Dong et al. (2020, doi:10.1029/2020EA001196) showed, the cloud seeding effect is distinct for adding IN particles when the cloud are supercooled liquid phase clouds. If the clouds are warm clouds, dust cannot affect the clouds by serving as IN, while they can modify cloud properties by serving as CCN. Thus, I would recommend the authors provide the information of cloud properties before the dust aerosols arrive (along with the cloud properties after the dust aerosols arrive).

Given the relative low altitude of the measurement site (1051 m asl) and considering that it is a tropical region, clouds are in the warm phase. The cloud phase found in this site has been included in the site description (section 2.1 Sampling Sites). In addition, the cloud microphysical properties, i.e. number concentrations, LWC and MVD, prior to the arrival of the air masses, are very clearly shown in Figs. 5&6 and discussed in the text.

Section 3 and 4, I wonder how the cloud droplet size distribution (since there are aircraft observations) varies with the dust amount.

We refer to our response in the previous reviewer comment. This is discussed already in great detail in the text and the size distributions are shown in Figs. 5 & 6.

Section 4. I love this discussion section about the potential causes for the observed findings. However, I do not feel very confident about the conclusions the authors obtained, particularly considering the highly complicated aerosol-cloud interactions with various influential factors. I wonder if the authors could provide more robust evidence. If not, I would suggest the authors weaken their tone to the conclusions.

The type of robust evidence that the reviewer request, while desirable, is rarely available in any aerosol-cloud interaction experiment because to prove beyond a doubt that cloud microphysical properties are due to specific aerosol properties, and environmental dynamics, the experiment would need not only an analysis of droplet residuals but also measurements of aerosols at their source. Then measurements would have to be made in a Lagrangian fashion to take into account aging, removal, etc.  If the reviewer can provide references to such studies, studies that don't use the type of scientific arguments that we have used in our analysis, we would welcome them. Otherwise, our approach of laying out the most probable mechanisms for forming and evolving clouds, followed with reasonable arguments based on the available information, is one that is most often used in studies published by other groups but in other locations. We see no need to weaken our conclusions as we have not made any statements that cannot be supported by the evidence.

The paper reports on three summers of measurements of aerosol optical properties and nss calcium at a coastal site (CSJ) and cloud properties at a mountain site (PDE) on Puerto Rico. These data are coupled with HYSPLIT back trajectories to assess the impacts of aerosols transported across the Atlantic on cloud properties and evolution at PDE. The bulk of the conclusions relies on meteorological parameters derived along the trajectories that are based on meteorological fields used when the HYSPLIT model is run. The end result is a study with provocative but speculative conclusions. The paper should be published as it provides information on the role of aerosol properties vs. atmospheric dynamics in cloud formation and evolution. It would be helpful to include a more detailed discussion of a follow-up study that could validate or inform the results presented here.

Lines 124 – 127: The accuracy of the position of the trajectories is discussed but, other than a brief mention of deriving precipitation from in situ and satellite observations (Lines 480 – 482), no mention is made of the uncertainty in deriving meteorological parameters along the trajectories. It would be helpful to add this information.

Meteorological data used in the HYSPLIT model comes from the Global Data Assimilation System (GDAS). GDAS adds several observations into a gridded 3-D model for forecasting weather using observations, which includes surface observations, weather balloons, ocean buoys, aircraft measurements, radar, and satellite observations. Several observation system simulation experiments have been carried out and have found that the GDAS output is reasonably comparable to nature runs and observations (eg. Kleist and Ide, 2015; Kren et al., 2020; Rangsanseri et al., 2020).

The above paragraph has been included in section 2.3 Classification of Sampling Periods

Figure 2: Please state in the caption what the black horizontal lines (solid and dashed) represent.

 Included.

Figure 3: Is the nss calcium shown in the figure in the aerosol or cloud phase? Please clarify in the caption.

 Included

Table 3: Please clarify exactly what sampling periods are represented in the table in the caption. Are these statistics for the combined low and high dust periods?

This should be Table 1. It is the statistics for the whole sampling period. The sampling period dates are described in section 2.2 Sampling Campaigns. A clarification has been included in the table caption.

Figure 4: What are the units of accumulated precipitation?

mm

Units have been added to the caption description

Line 294: Please add "size distributions of CLOUD DROP number concentration" for clarification.

Included

Line 389: What is meant here by "cloud concentration"? Cloud drop concentration? Cloud fraction?

Cloud droplet concentration. This has been corrected.

Line 421: Should this be "….acting as cloud condensation nuclei and thus BE THE MAIN AEROSOL COMPONENT responsible…"?

Corrected.

Lines 555 – 558: The conclusions are, indeed, highly speculative. What additional information is required to reach conclusions with more certainty? It would be helpful to have that information rather than the generic statement that "…a much more detailed and long-term measurement program…" is required. What measurements should be included both at PDE and CSJ? Add more specifics about the required cloud and chemistry models needed to validate the results presented here.

Clearly, the conclusions posted here must be somewhat speculative in nature and require a much more detailed and long-term measurement program, coupled with cloud and chemistry models to validate these speculations. Nevertheless, the results are consistent with previous studies and, most importantly, provide a well-documented set of measurements to enhance the current data set of similar observations in the Caribbean.

Currently in progress is the next step to extend this study with a more comprehensive suite of sensors over a period of time that covers several years in order to take into account the year-to-year variability in synoptic and mesoscale weather patterns that modify the trajectories of dust and pollution transported to the Caribbean. Following the passage of Hurricane Marie, in September 2017, two years after the current study ended, both the mountain and coastal research sites were destroyed with all the equipment. As a result, through funding from the National Science Foundation, these sites have been rebuilt and new cloud and aerosol instrumentation purchased and installed. This new facility will be fully on-line in October, 2021 and begin long-term measurements that will expand on the results that have been reported here. These new measurements, along with simulations with the Weather Research Forecast (WRF) will directly address the questions raised in the current study and begin moving the conclusions from the realm of speculation to statistically supported facts.

---

## Referee Report (RR1)

Review of : acp-2021-88

Measurement Report: Impact of African Aerosol Particles on Cloud Evolution in a Tropical Montane Cloud Forest in the Caribbean
authors: Elvis Torres-Delgado1, Darrel Baumgardner2, Olga L. Mayol-Bracero1

Comments: this manuscript describes a valuable dataset and as such should be published. It does need some further polish to be acceptable for publication.

Introduction:
Line 29: the authors mention the activity of dust as a CCN is less clear. Some studies the authors may also wish to consider in this vein are Twohy et al., 2009; Denjean et al., 2015 (which suggests the CCN activity of dust is primarily through its size, based on Puerto Rico measurements; the 3rd author is also part of this author list); Edwards et al., 2021, which examined the CCN activity of dust reaching Miami, Florida (and found it minor compared to smoke).  References within these papers can also help the authors expand their literature review.

Line 31: "later" -> "latter"

Lines 64 & line 73: some repetition here.

Lines 83-84: here we learn that the sampling campaigns were of widely differing lengths. It would be worth including the # of days contained within each campaign within the abstract. Something on the climatology on when dust is encountered in Puerto Rico would be nice to see as well, here or elsewhere. Data for Barbados and Miami are available from Zuidema et al 2019.

Table 1, p 8: include the number of days or hours included in the statistical values for each year.

Line 234: how is the Sahel differentiated from the Sahara? Is there a latitude line that is invoked near the coastline at about 18W - or further inland? fig. 4b doesn't make clear as the two distributions overlap completely. What is the significance of distinguishing these 2 populations? I might think that air from the Sahel contains biomass-burning aerosol, whereas that from the Sahara doesn't, but the authors do not discuss this.

Fig 5, 6: I would suggest replotting these simply as 'hours since air mass arrival', as there is nothing of meaning in the diurnal cycle according to lines 323-324.

Discussion: it's worth mentioning somewhere that the changes in cloud properties also depend on what the properties were before a new air mass moved in. How important of an effect do the authors believe this to be?

P. 19: I would suggest placing the discussion on the diurnal cycle earlier, rather than here, as it is relevant to figs 5 and 6.

Data availability: the data should be made publicly available through a data repository and be associated with a digital object identifier.

References:

Twohy, C. H., et al. (2009), Saharan dust particles nucleate droplets in eastern Atlantic clouds, Geophys. Res. Lett., 36, L01807, doi:10.1029/2008GL035846.

Denjean, C., S. Caquineau, K. Desboeufs, B. Laurent, M. Maille, M. Quiñones Rosado, P. Vallejo, O. L. Mayol-Bracero, and P. Formenti (2015), Long-range transport across the Atlantic in summertime does not enhance the hygroscopicity of African mineral dust, Geophys. Res. Lett., 42, 7835–7843, doi:10.1002/2015GL065693.

Edwards, E.-L., A. Corral, H. Dadashazar, A. Barkley, C. Gaston, P. Zuidema, A. Sorooshian, 2021: Impact of various air mass types on cloud condensation nuclei concentrations along coastal southeast Florida. *Atmos. Env.*, doi:10.1016/j.atmosenv.2021.118371

Zuidema, P., C. Alvarez, S. J. Kramer, L. Custals, M. Izaguirre, P. Sealy, J. M. Prospero, E. Blades, 2019: Is summer African dust arriving earlier to Barbados? The updated long-term in-situ dust mass concentration records from Ragged Point, Barbados and Miami, Florida. *Bull. Am. Meteor. Soc.*, **100**, p. 1981-1986, doi:BAMS-D-18-0083.1

---

## Author Response (AR2)

Torres-Delgado et al. "Measurement Report: Impact of African Aerosol Particles on Cloud Evolution in a Tropical Montane Cloud Forest in the Caribbean"

Answer to reviewer's reports

Line 16-18, I might be asking a stupid question, but I really want to learn: how did the authors compare/quantify the relative importance of meteorology and transported aerosols on cloud processes? In my understanding, if there are no aerosols, there are no clouds; if there are no supersaturation condition, there are no clouds. How could we determine their relative importance considering that they are totally different variables with different units? May the authors help explain more?

We understand the reviewer's question and already in the manuscript we have explained that we are unable to quantify the relative importance of meteorology compared to aerosols without an additional modeling effort.

Line 102-106, How consistent are the LWC from BCP and hotwire?

The authors have not seen a study that compares both techniques for retrieving LWC. However, our results compared well with the results from other studies in the same area that used optical measurements for retrieving information on cloud properties.

Line 219-221, Description about the results shown in Table 1 is necessary here.

The authors decided to eliminate this table as it was not used in the rest of the manuscript.

Figure 4. Unit for latitude and longitude is necessary. Also, the word in the figure is too blurry.

The figure has been replaced.

Line 305, LWC has been defined earlier in the data section.

Edited.

Figure 9, the unit of temperature seems not right. There is spell error for "relative humidity"

The unit temperature is correct. The typo was corrected.

Review of : acp-2021-88

Measurement Report: Impact of African Aerosol Particles on Cloud Evolution in a Tropical Montane Cloud Forest in the Caribbean

authors: Elvis Torres-Delgado1, Darrel Baumgardner2, Olga L. Mayol-Bracero1

Comments: this manuscript describes a valuable dataset and as such should be published. It does need some further polish to be acceptable for publication.

Introduction:

Line 29: the authors mention the activity of dust as a CCN is less clear. Some studies the authors may also wish to consider in this vein are Twohy et al., 2009; Denjean et al., 2015 (which suggests the CCN activity of dust is primarily through its size, based on Puerto Rico measurements; the 3rd author is also part of this author list); Edwards et al., 2021, which examined the CCN activity of dust reaching Miami, Florida (and found it minor compared to smoke). References within these papers can also help the authors expand their literature review.

The pertinent references have been included along with a brief explanation of the findings. Indeed, they improved the literature review.

Line 31: "later" -> "latter"

Edited.

Lines 64 & line 73: some repetition here.

These lines were edited to make it easier to read.

Lines 83-84: here we learn that the sampling campaigns were of widely differing lengths. It would be worth including the # of days contained within each campaign within the abstract.

Included.

Something on the climatology on when dust is encountered in Puerto Rico would be nice to see as well, here or elsewhere. Data for Barbados and Miami are available from Zuidema et al 2019.

We are working on a separate manuscript that includes a long-term record of dust properties measured in Puerto Rico.

Table 1, p 8: include the number of days or hours included in the statistical values for each year.

The authors decided to eliminate this table as it was not used in the rest of the manuscript.

Line 234: how is the Sahel differentiated from the Sahara? Is there a latitude line that is invoked near the coastline at about 18W - or further inland? fig. 4b doesn't make clear as the two distributions overlap completely. What is the significance of distinguishing these 2 populations? I might think that air from the Sahel contains biomass-burning aerosol, whereas that from the Sahara doesn't, but the authors do not discuss this.

Plot 4b shows the altitude of the air masses. Plot 4a shows the origin of the air masses. From Plot 4a we can distinguish the different air mass influences.

Fig 5, 6: I would suggest replotting these simply as 'hours since air mass arrival', as there is nothing of meaning in the diurnal cycle according to lines 323-324.

The authors understand that it is important to illustrate that the clouds that formed from different air masses did so at different times of the day. For this reason, the authors decided not to incorporate the suggested change.

Discussion: it's worth mentioning somewhere that the changes in cloud properties also depend on what the properties were before a new air mass moved in. How important of an effect do the authors believe this to be?

In this study we did not find evidence that the cloud properties are being affected by what the conditions were before. On line 188 it is mentioned that there were no trends in the cloud optical properties that could be link to diurnal variations or local emissions.

P. 19: I would suggest placing the discussion on the diurnal cycle earlier, rather than here, as it

is relevant to figs 5 and 6.

The authors understand that it is important to have the discussion of the four different air mass histories, as the diurnal cycle is discussed under this scope. Therefore, we decided to leave this section where it is.

Data availability: the data should be made publicly available through a data repository and be

associated with a digital object identifier.

The authors prefer to keep data availability upon request.

---

## Author Response (AR3)

Response to editor comments:

Please put the data in a public repository for the access and add a location under the "Data Availability" (this is also requested by one reviewer). This will increase the impacts of your study.

The data has been included in a data repository and the link to it has been included in the manuscript. Also, it has been included here: https://osf.io/a42fj/?view_only=d6c2a964e8a74b7c8e721dabd5cd9840

Currently you are using the terminology of Ice nuclei (IN). Please use Ice nucleating particles (INP), as this is more used term.

The term ice nuclei has been replaced with ice nucleating particle.

A previous email from the editorial support team requested that we included a copyright logo on the Google Earth map we present in Fig. 1. This has been included.